# STING inhibition enables efficient plasmid-based gene expression in primary vascular cells: A simple and cost-effective transfection protocol

**Shuai Yuan**[1,2]*, **Adam C. Straub**[1,2,3]*

**1** Department of Pharmacology and Chemical Biology, University of Pittsburgh, Pittsburgh, PA, United States of America, **2** Heart, Lung, Blood and Vascular Medicine Institute, University of Pittsburgh, Pittsburgh, Pennsylvania, United States of America, **3** Center for Microvascular Research, University of Pittsburgh, Pittsburgh, Pennsylvania, United States of America

* yuans@pitt.edu (SY); astraub@pitt.edu (ACS)

**Data Availability Statement:** All statistical analyses and data plotting were conducted in R. The R markdown files and original data, including the original western blot images, have been uploaded

## Abstract

Plasmid transfection in cells is widely employed to express exogenous proteins, offering valuable mechanistic insight into their function(s). However, plasmid transfection efficiency in primary vascular endothelial cells (ECs) and smooth muscle cells (SMCs) is restricted with lipid-based transfection reagents such as Lipofectamine. The STING pathway, activated by foreign DNA in the cytosol, prevents foreign gene expression and induces DNA degradation. To address this, we explored the potential of STING inhibitors on the impact of plasmid expression in primary ECs and SMCs. Primary human aortic endothelial cells (HAECs) were transfected with a bicistronic plasmid expressing cytochrome b5 reductase 4 (CYB5R4) and enhanced green fluorescent protein (EGFP) using Lipofectamine 3000. Two STING inhibitors, MRT67307 and BX795, were added during transfection and overnight post-transfection. As a result, MRT67307 significantly enhanced CYB5R4 and EGFP expression, even 24 hours after its removal. In comparison, MRT67307 pretreatment did not affect transfection, suggesting the inhibitor's effect was readily reversible. The phosphorylation of endothelial nitric oxide synthase (eNOS) at Serine 1177 (S1177) by vascular endothelial growth factor is essential for endothelial proliferation, migration, and survival. Using the same protocol, we transfected wild-type and phosphorylation-incapable mutant (S1177A) eNOS in HAECs. Both forms of eNOS localized on the plasma membrane, but only the wild-type eNOS was phosphorylated by vascular endothelial growth factor treatment, indicating normal functionality of overexpressed proteins. MRT67307 and BX795 also improved plasmid expression in human and rat aortic SMCs. In conclusion, this study presents a modification enabling efficient plasmid transfection in primary vascular ECs and SMCs, offering a favorable approach to studying protein function(s) in these cell types, with potential implications for other primary cell types that are challenging to transfect.

to a public repository (10.6084/m9.figshare.
24982932) and can be used to reproduce figures in
this paper.

**Funding:** Financial support for this work was
provided by the National Institutes of Health (www.
nih.gov) R35 HL161177 (A.C.S.), National
Institutes of Health R01 HL 153532 (A.C.S.), and
American Heart Association (www.heart.org)
Established Investigator Award 19EIA34770095 (A.
C.S.). Funders did not play any role in the study
design, data collection and analysis, decision to
publish, or preparation of the manuscript.

**Competing interests:** NO authors have competing
interests Enter: The authors have declared that no
competing interests exist.

# Introduction

The introduction and subsequent expression of exogenous genes into eukaryotic cells, commonly known as transfection, stands as a foundational technique in molecular cell biology. Transfection enables elucidation of gene function, characterization of protein interactions, synthesis of proteins, and production of viruses. However, the success of these ventures hinges on the effectiveness of the transfection method employed. Various reagents and techniques have been developed and fine-tuned to introduce DNA into cells. Commonly used transfection reagents include calcium phosphate, cationic polymers (i.e., polyethylenimine), and liposome-based reagents (i.e., Lipofectamine) [1–3]. Unfortunately, transfecting primary vascular endothelial and smooth muscle cells has been challenging with these reagents [4–6]. Higher transfection efficiency in these cell types can be achieved with more complicated electroporation methods at the expense of increased cytotoxicity, which makes it unreliable to investigate signaling pathways in a physiological context [7, 8]. Furthermore, viral-based methods allow for efficient transduction, while customizing viruses requires expert knowledge and can be labor, time, and cost-limiting.

Although successful cellular entrance is necessary for a transfection experiment, it is not the only limiting factor of transgene expression in the host cells. After entering the cell, the transgene-carrying vector activates the stimulator of interferon genes (STING) pathway as an innate immune response to cytosolic DNA [9]. The downstream upregulation of interferons and interferon-stimulated genes, mediated by TANK-binding kinase 1 (TBK1) and I-kappa-B kinase epsilon (IKKε), induces expression of the $2'$-$5'$-oligoadenylate synthetase (OAS) family genes [10–12]. OAS activates endoribonuclease and RNase L, which can suppress transgene expression. Indeed, it has been shown that inhibitors against TBK1 and IKKε effectively suppressed the DNA transfection-induced interferon response and enhanced transgene expression in multiple cell lines and primary human T cells [13]. In this study, we examine whether two TBK1/IKKε inhibitors, MRT67307 and BX795, improve transgene expression in DNA-transfected primary vascular cells. We further demonstrate a Lipofectamine-based transfection protocol optimized for primary endothelial and smooth muscle cells, with an example of its application in studying vascular endothelial growth factor (VEGF) signaling.

# Materials and methods

## Cell culture

Human aortic endothelial cells (HAEC), human aortic smooth muscle cells (HASMC), and rat aortic smooth muscle cells (RASMC) were purchased from Lonza. Cells were maintained in endothelial (EGM-2, Lonza, CC-3162) or smooth muscle growth medium (SmGM-2, Lonza, CC-3182) with 5% $CO_2$ at 37˚C. We received primary cells at Passage 3 and used them in experiments for population doubling no more than 13 times.

## Molecular cloning

A lentiviral empty cloning vector, pCIG3, was obtained from Addgene (Plasmid #78264). The subclone we generated from the Addgene stock showed little promoter activity, so we substituted its minimal CMV promoter with the CMV enhancer and promoter region from pcDNA3.1 (Invitrogen). We also cloned the lentiviral genome between 5' and 3' LTRs onto a pLenti backbone and inserted a more flexible multiple cloning sequence. The final lentiviral cloning product, named pCIGX, was confirmed using whole plasmid sequencing (Genewiz, Plasmid-EZ).

The plasmids with coding sequences of human cytochrome b5 reductase 4 (CYB5R4) and human nitric oxide synthase 3 (eNOS) were purchased from Transomic Technologies (Clone ID BC063294, BC025380). The S1177A mutation of eNOS was achieved by PCR mutagenesis using Q5 DNA polymerase (NEB, M0491S). All coding sequences were confirmed by Sanger sequencing (Genewiz).

### Plasmid transfection

Transfection was performed using Lipofectamine 3000 according to the manufacturer's instructions with modifications. The optimal transfection protocols vary between cell types. For HAECs, 20,000–23,000 cells/cm$^2$ were seeded for the day before transfection unless specified otherwise. The goal was to achieve 80–90% confluence at the time of transfection. For each well of a 12-well plate, 0.2 or 0.4 μg DNA was diluted in 100 μl Opti-MEM and mixed with P3000 in the 1:2 mass ratio, while 1.5 μl Lipofectamine 3000 was diluted in another 100 μl Opti-MEM. The diluted DNA and Lipofectamine 3000 were mixed by vortexing and incubated at room temperature for 10 minutes. Afterward, the lipid-DNA complex was added to cells. HAECs were typically incubated with transfection reagents for up to 4 hours before the medium was replenished. However, in case of significant cell stress or death, transfection was stopped early.

For HASMCs and RASMCs, cells were seeded at 15,000 cells/cm$^2$ density so that they reached 70% confluence. Transfections were set up similarly to the above, except that 0.4 μg DNA was used for each well of a 12-well plate, and the cells were incubated with the lipid-DNA complex overnight.

MRT67307 and BX795 were added to the cell culture medium to achieve the various concentrations for the different periods. During the experiment, all cell types were cultured in fully supplemented EGM-2 or SmGM-2. It was not necessary to remove serum, growth factors, or antibiotics.

### Immunoblotting

Cells were lysed in 2x Laemmli buffer with 5% β-mercaptopyruvate and heated at 95˚C for 10 minutes. Protein samples were resolved on NuPAGE Bis-Tris 4–12% gels and transferred to nitrocellulose membranes. Membranes were blocked in phosphate-buffered saline (PBS) with 3% bovine serum albumin (BSA) and 0.05% Tween-20 at room temperature for 1 hour, followed by overnight incubation with primary antibodies against CYB5R4 (Santa Cruz, sc-390569, 1:1000), α-Tubulin (Sigma, T6074, 1:10000), GFP (Abcam, ab6673, 1:1000), eNOS (BD, 610296, 1:1000), or eNOS phosphorylated at S1177 (Cell Signaling, 9571, 1:1000). Primary antibodies were then probed by IRDye 680RD or 800CW-conjugated secondary antibodies (Licor, 1:10000) and imaged with a Licor Imager. Band intensities were quantified using Licor Image Studio and normalized to α-Tubulin. Results were reported as fold increases over the control group.

### Immunostaining

HAECs were seeded on glass coverslips coated with 0.5% gelatin. At the time of collection, cells were fixed with 4% paraformaldehyde and permeabilized with 0.5% Triton X-100. Unspecific antibody binding sites were blocked by PBS with 10% horse serum, 1% BSA, and 0.1% Tween-20. Coverslips were incubated with antibodies against GFP, eNOS, and eNOS phosphorylated at S1177 overnight at 4˚C. Primary antibodies were detected with fluorophore-conjugated secondary antibodies. Images were captured using a Nikon A1 confocal microscope.

## Statistics

All experiments were repeated on at least 3 different days, and each repeat was considered a biological replicate (n = 1). One-way analysis of variance (ANOVA) Tukey's post-hoc tests were performed in the R environment.

## Results

### STING inhibition enhances plasmid gene expression

Cytochrome b5 reductases are a family of proteins crucial for maintaining lipid metabolism, hemeprotein redox state, and cellular redox regulation [14–17]. They are ubiquitously expressed in mammalian cell types at various levels [18]. Cytochrome b5 reductase 4 (CYB5R4) is expressed in human endothelial cells with relatively low RNA abundance [19]. We cloned CYB5R4 in a mammalian expression vector (pCIGX) to examine the expression efficiency of an endogenous gene with plasmid transfection. Meanwhile, the pCIGX-CYB5R4 co-expresses green fluorescence protein (EGFP), providing an exogenous gene as a secondary readout.

To test whether MRT67307 or BX795 affect plasmid gene expression, we treated cultured HAECs with either inhibitor at various concentrations immediately before the addition of the lipid-DNA complex. After 4-hour incubation, cells were thoroughly rinsed to remove transfection reagents, while inhibitors were replenished in fresh media. 24 hours after the medium change, cells were lysed to measure CYB5R4 and GFP expressions (Fig 1). As expected, the standard lipofectamine transfection (0 μM MRT67307 or BX795) did not result in significant over-expression of CYB5R4 or GFP. In comparison, MRT67307 at 5 μM concentration increased both CYB5R4 (4 folds) and GFP (5 folds) expressions. BX795 exhibited similar dose-dependent enhancements of plasmid gene expression. However, its effects were not as consistent as MRT67307, potentially due to toxic effects.

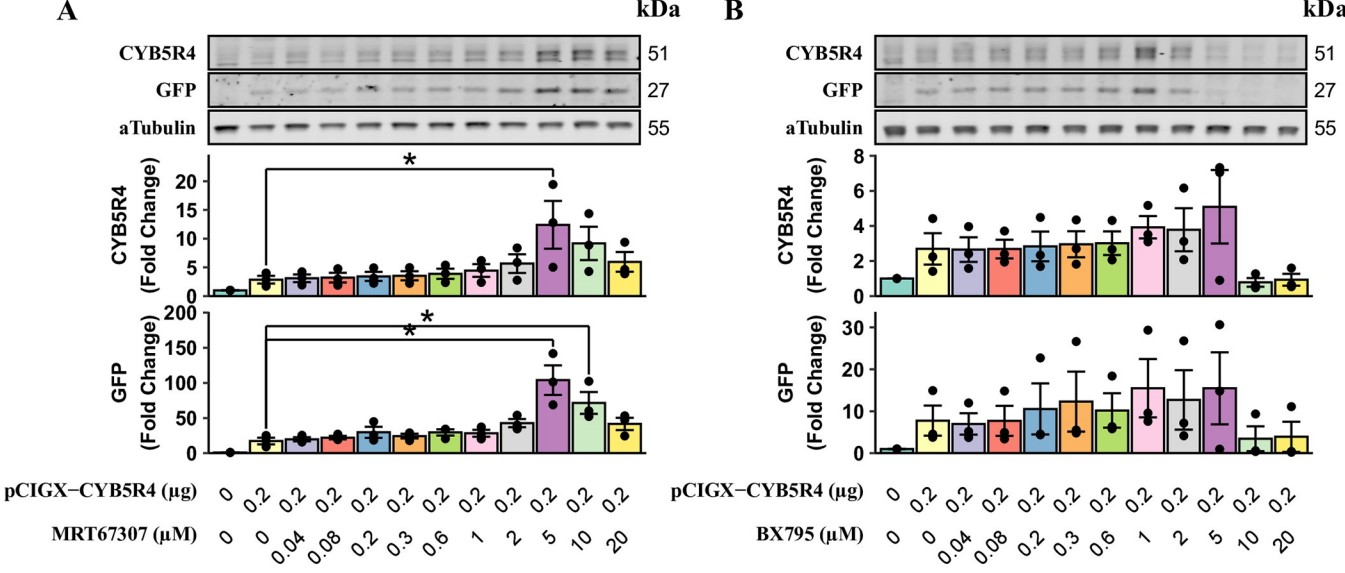

**Fig 1. MRT67307 and BX795 enhance plasmid gene expression in a concentration-dependent manner.** HAECs in 12-well plates were treated with the same amount of Lipofectamine but different amounts of plasmid DNA (0 or 0.2 μg pCIGX-CYB5R4) for 4 hours. Cells receiving plasmid DNA transfection were incubated with MRT67307 (A) or BX795 (B) at the indicated concentrations for the duration of transfection and the first 24 hours after transfection. Protein expression was measured using western blotting, and the group with 0.2 μg DNA and 0 μM inhibitor was set as the baseline to calculate fold change. * indicates a significant difference between two connected groups (n = 3, p < 0.05).

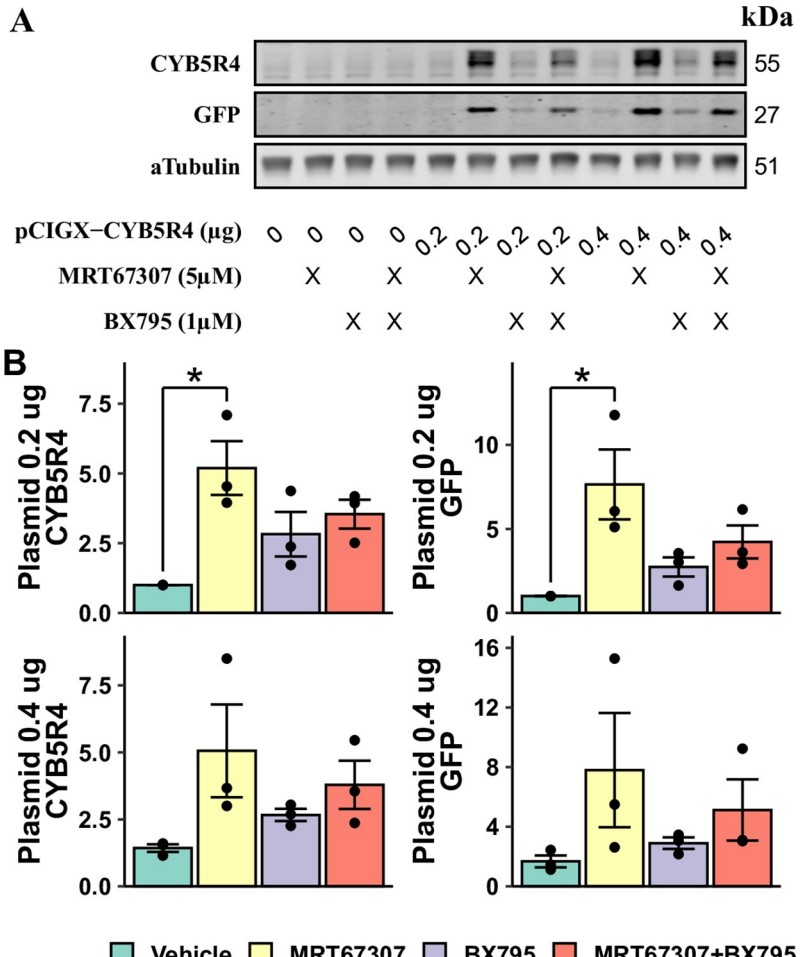

**Fig 2. Optimization of inhibitor treatment and plasmid concentration.** A well of a 12-well plate HAECs were transfected with 0, 0.2, or 0.4 μg pCIGX-CYB5R4 and treated with MRT67307 or BX795. The western blot image of a biological replicate is shown in (A). Protein expression levels are normalized to the group with 0.2 μg plasmid DNA and no inhibitor treatment (B). Significant changes are annotated by * between connected groups (n = 3, p < 0.05).

### Further optimization of the transfection protocol

We aimed to improve plasmid gene expression further with two strategies: 1) increasing the plasmid load and 2) combining MRT67307 and BX795 at sub-toxic concentrations for a more complete inhibition of the STING pathway. With these adjustments, HAECs were transfected with pCIGX-CYB5R4, the same as above. However, neither condition showed higher CYB5R4 or GFP expression than 0.2 μg plasmid (per ~80,000 cells) with 5 μM MRT67307 (Fig 2). Instead, the addition of BX795 adversely affected plasmid gene expression. Therefore, the rest of the transfection experiments in HAECs were performed only using MRT67307.

### Prolonged exposure to MRT67307 did not further improve plasmid gene expression

As inhibitor treatments for extended periods of time may cause cytotoxicity, we sought to determine the minimum treatment time for MRT67307 to enhance plasmid gene expression. In this experiment, we performed 4 hour lipofectamine transfections in HAECs while exposing

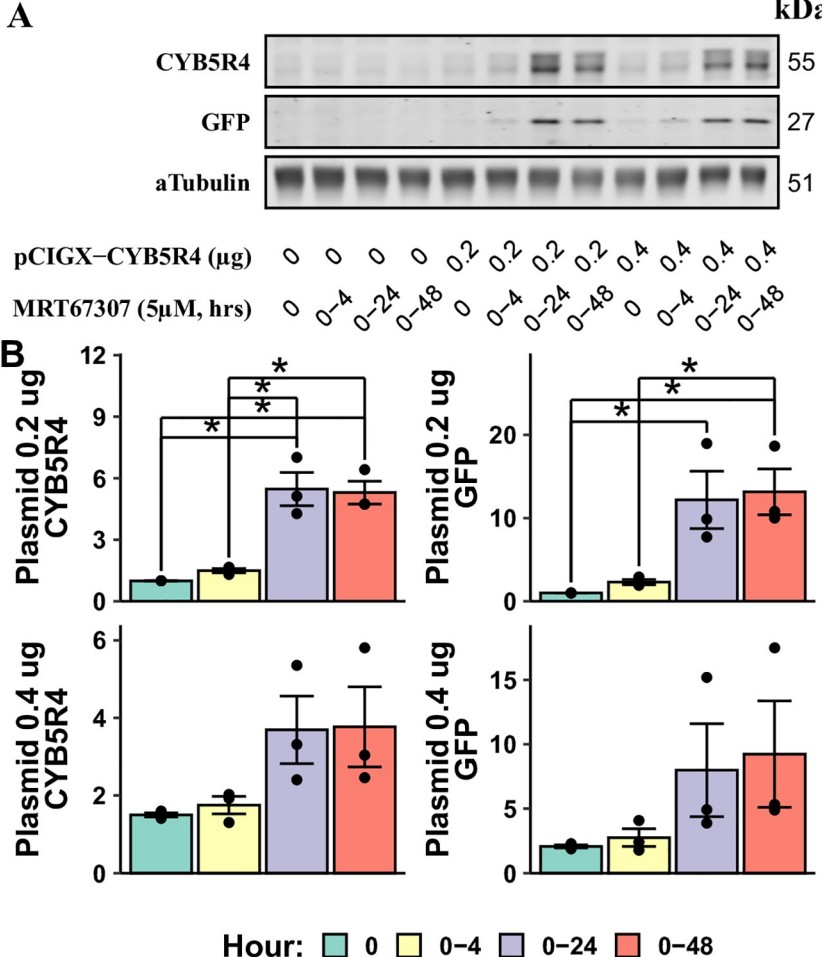

**Fig 3. Effects of MRT67307 exposure time on transgene expression.** HAECs were transfected with different amounts of pCIGX-CYB5R4 in 12-well plates. From the start of transfection (0 h), cells were treated with 5 μM MRT67307 for the indicated time frames. (A) shows a representative immunoblotting experiment to measure transgene expression. Results are presented as fold change over the group with 0.2 μg DNA and 0 μM MRT67307 (B). * annotates significant differences between indicated groups (n = 3, p < 0.05).

cells to MRT67307 for up to 48 hours. All cells were collected at the 48-hour time point to measure protein expressions (Fig 3). The inclusion of MRT67307 only during transfection (0–4 hours) failed to increase CYB5R4 or GFP expressions. Meanwhile, the MRT67307 treatment beyond the first 24 hours (0–48 hours) provided no more improvement compared to the 0–24 hour group. Increasing plasmid load led to inconsistent over-expressions, potentially due to cytotoxicity.

## The effect of MRT67307 is transient and reversible

As MRT67307 is only required for the first 24 hours to allow plasmid gene expression, it remains unclear whether its inhibition of the STING pathway sustains after its removal. To test the reversibility of MRT67407, we seeded HAECs at a lower density in 12-well plates (18,000 cells/cm$^2$), which was considered the 0-hour time point. In the following 3 days, three groups of cells were exposed to 5 μM MRT67307 for only the indicated intervals (0–24, 24–48, or 48–72 hours). All groups received medium changes each day to add or wash out

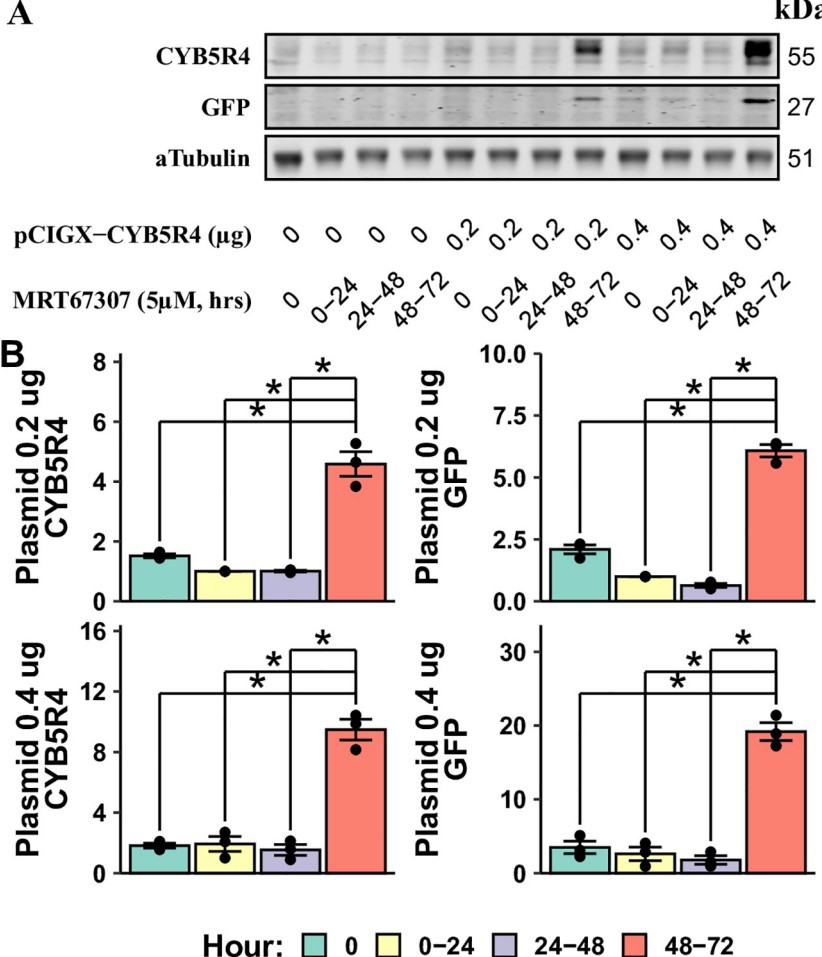

**Fig 4. The effect of MRT67307 is reversible, while transgene expression is retained.** HAECs were plated in 12-well plates (0 h time point) and treated with 5 μM MRT67307 for 24 hours at the indicated time frames. The inhibitor was washed off after the 24-hour treatment, accompanied by medium change for all groups. After wash-off for the 24–48-hour group, all cells received 4-hour MRT67307 treatment and lipofectamine transfection with 0, 0.2, or 0.4 μg pCIGX-CYB5R4. A representative western blot is shown in (A). Only the group exposed to MRT67307 before and after transfection (48–72 h group) shows increased transgene expression (B) (* for $p < 0.05$, n = 3).

MRT67307. Cells were transfected with pCIGX-CYB5R4 with Lipofectamine for 4 hours at the 48-hour time point and collected at the 72-hour time point to quantify CYB5R4 and GFP expressions using western blotting (Fig 4). Consistent with the above, cells received MRT67307 treatment during transfection and within the first 24 hours after transfection (the 48–72 hour group) exhibited robust expression of both CYB5R4 and GFP. In contrast, cells with MRT67307 washed out 24 hours before transfection (the 0–24 hour group) or immediately before transfection (the 24–48 hour group) showed no protein over-expression at all. Unlike previous experiments (Figs 2 and 3), in this set of experiments, the higher DNA amount during transfection doubled the expression levels of both CYB5R4 and GFP.

## Using the modified transfection protocol in a functional study

Next, we wanted to examine whether the transfection-induced protein expression is functionally intact. Nitric oxide synthase 3 (eNOS) is the predominant source of nitric oxide in endothelial cells [20]. It is well documented that vascular endothelial growth factor (VEGF)

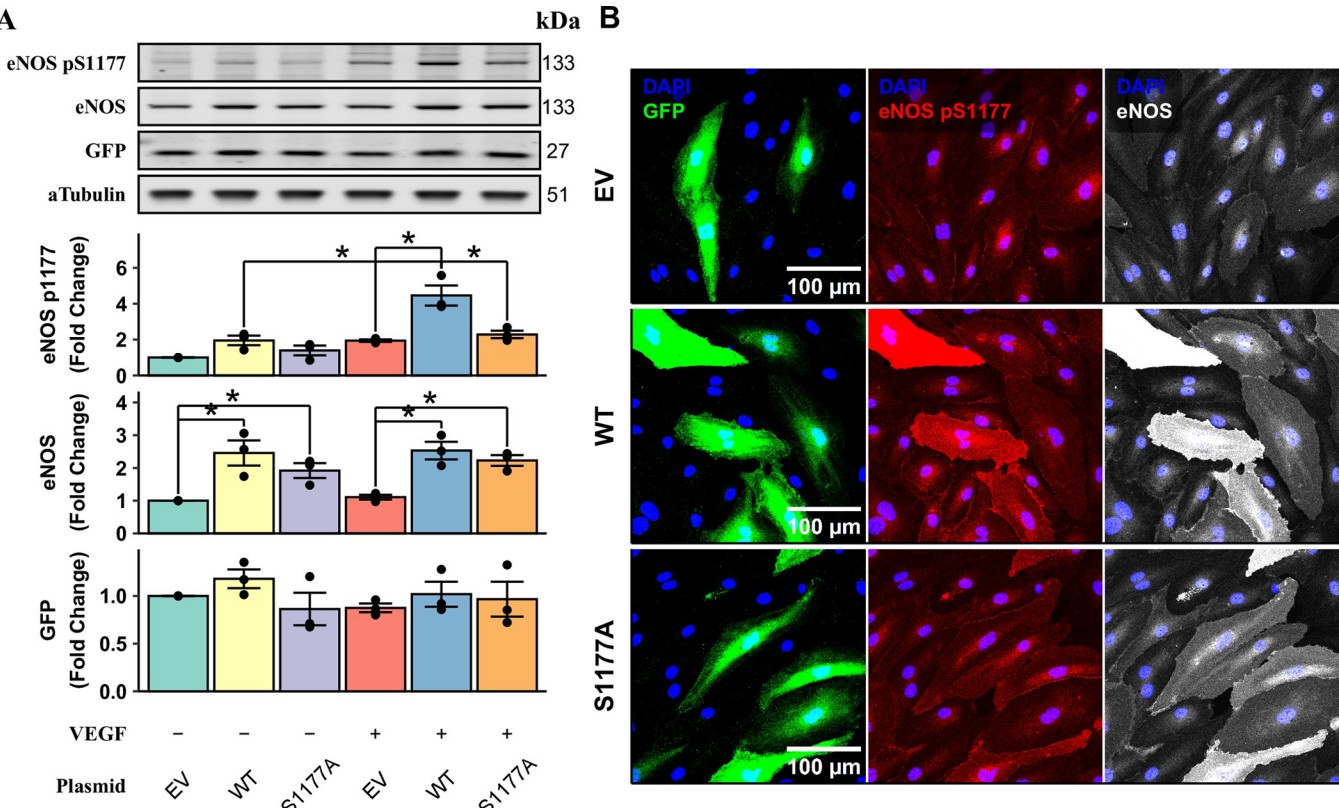

**Fig 5. Over-expressed eNOS residues in the physiological compartment and responses to VEGF signaling.** HAECs were transfected with 0.2 μg pCIGX plasmids carrying no transgene (EV), wild-type eNOS (WT), or S1177A mutant eNOS (S1177A) with optimized MRT67307 treatment. 48 hours after transfection, cells were deprived of serum and growth factors for 2 hours and treated with 20 ng/ml VEGF for 5 minutes. Western blot showed WT but not S1177A eNOS is phosphorylated by VEGF treatment (A) (* annotates $p < 0.05$, n = 3). Immunostaining shows over-expressed eNOS in GFP-positive cells co-localized with the plasma membrane and ER (B). WT but not S1177A eNOS over-expressing cells show increased phosphorylation of eNOS S1177.

phosphorylates human eNOS at Serine 1177 (S1177) and activates the enzyme essential for endothelial cell proliferation, migration, and survival [21–23]. Wild-type (WT) and phosphorylation-incapable mutant (S1177A) eNOS were cloned into the pCIGX vector. HAECs were transfected with 0.2 μg DNA per 80,000 cells for 4 hours in the presence of 5 μM MRT67307. MRT67307 was kept in the medium for the first 24 hours and withdrew for another day. Cells were then starved in endothelial basal medium for 2 hours and treated with 20 ng/ml VEGF for 5 minutes. Western blotting shows 2-fold increase of WT or S1177A eNOS compared to the empty vector (EV) (Fig 5A). GFP expressions were similar between groups, indicating equal plasmid DNA load (Fig 5A). Notably, VEGF-induced eNOS phosphorylation at S1177 was only elevated in cells over-expressing WT but not S1177A eNOS (Fig 5A). Over-expressed WT and S1177A eNOS in GFP-positive cells were correctly localized with plasma membrane (Fig 5B). Similarly, cells transfected with S1177A eNOS showed similar levels of S1177-phosphorylated eNOS compared to GFP-negative cells (Fig 5B).

## STING inhibition allows plasmid gene expression in vascular smooth muscle cells

Vascular smooth muscle cells are also difficult to transfect with plasmid DNA. We tested the effects of MRT67307 and BX795 in human (HASMC) and rat (RASMC) aortic smooth muscle cells. Cells were seeded in 12-well plates at 15,000 cells/cm$^2$. Transfection of pCIGX-CYB5R4

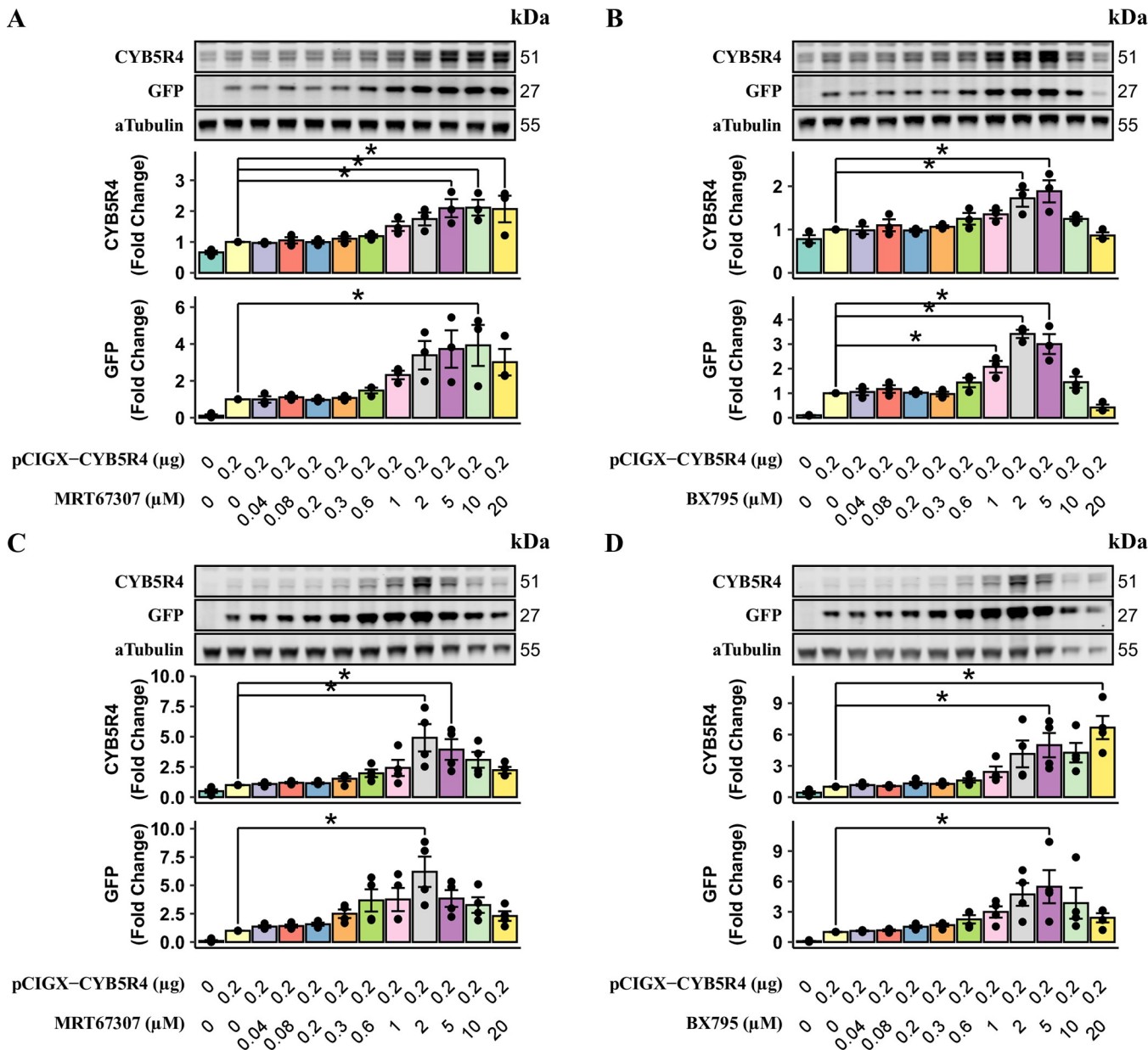

**Fig 6. MRT67307 and BX795 enhance transgene expression in vascular smooth muscle cells.** HASMCs (A and B) and RASMCs (C and D) were transfected with 0.2 μg pCIGX-CYB5R4 in 12 well plates in the presence of MRT67307 (A and C) and BX795 (B and D) at various concentrations. Protein expression levels are normalized to cells receiving 0.2 μg plasmid and 0 μM inhibitor. Statistical differences are indicated by * between two groups (n = 4, p < 0.05).

was performed the same way as for HAECs, except that cells were left with a lipid-DNA complex overnight. Both inhibitors significantly increased CYB5R4 and GFP expressions in HASMCs (Fig 6A and 6B) and RASMCs (Fig 6C and 6D), suggesting their sensitivity to MRT67307 and BX795 differed from those of HAECs (Fig 1).

## Discussion

The ability to interfere with a gene's expression, either up or down-regulation, is the foundation for establishing causal relationships between a gene and a phenotype. There is a myriad of

tools to achieve this goal. Nowadays, viral transduction is often the first choice as viral packaging systems become readily accessible and safe to use. Despite their popularity, caution is warranted to avoid critical caveats. Adenovirus can be acquired at high titers and results in high transduction efficiency. However, the E4 gene in the adenoviral genome activates strong Akt phosphorylation in endothelial cells, complicating the studies of endothelial proliferation, survival, and nitric oxide signaling [24]. Lentivirus actively integrates the cargo gene into the host genome, which may cause gene silencing, activation, and aberrant transcript splicing [25]. Even though lentiviral transduction does not require cell proliferation, primary cells, such as endothelial cells, usually need 48–72 hours to express the transgene effectively [26, 27], while plasmid transfection causes robust overexpression in 24 hours. Additionally, accurately tittering viruses can be challenging, especially when different transgenes affect viral packaging or the viability of cells used for tittering. In comparison, plasmid transfection is more straightforward and easily controlled.

Unfortunately, standard transfection methods are challenging when applied to primary vascular endothelial cells and smooth muscle cells. Previous studies have focused on searching for more effective (20–40% transfection positivity) approaches to introduce DNA into cells. For human umbilical vein endothelial cells (HUVECs), liposome-based transfection reagents result in better transfection than other cationic polymers and calcium phosphate [4, 5]. Higher transfection efficiency (~70%) was achieved in HUVECs, HAECs, and human microvascular endothelial cells (HMVEC) using electroporation, but cell survival rates were only 10–30% [7]. Additionally, the amounts of DNA used in these studies were 10–20 μg per million cells, which is not cost-efficient.

Our laboratory and others routinely perform siRNA transfection in primary endothelial cells with Lipofectamine 3000. We did not observe significant cell death after incubating cells with liposome complexes overnight, suggesting Lipofectamine alone was not cytotoxic under a standard protocol. However, when incubated with Lipofectamine-DNA complex overnight, we could not recover viable cells to examine plasmid gene expression. Similarly, it has been reported that Lipofectamine with DNA caused more endothelial cell death than Lipofectamine or DNA alone [4]. Therefore, we reasoned that the poor transfection efficiency in endothelial cells was not due to unsuccessful DNA delivery but impaired plasmid gene expression.

The presence of cytosolic DNA is sensed by cyclic GMP-AMP synthase (cGAS), which stimulates STING protein and downstream TBK1 and IKK kinases [28, 29]. The signaling complex leads to the activation of Type I Interferon response [30] and autophagosome recruitment [31–33]. As a part of the innate immune response, the STING pathway is essential for the sensing and clearance of viruses and other microorganisms. However, the STING pathway is also activated in plasmid DNA-transfected cells [34]. A seminal study by Fu Y and colleagues showed that plasmid gene expression was improved in cGAS or STING-deficient cell lines [13]. They also demonstrated that TBK1 inhibitors, MRT67307 and BX795, enhance transfection efficiency with Lipofectamine in multiple cell lines and primary T cells in human peripheral blood mononuclear cells with electroporation. In our study, we demonstrated the potential of STING inhibition in facilitating plasmid gene expression in primary vascular cells.

We constructed a plasmid co-expressing GFP and CYB5R4, a protein endogenously expressed in mammalian cells. In HAECs, the standard lipofectamine transfection protocol barely induced any transgene expression (Fig 1). With 5 μM MRT67307 treatment, CYB5R4 and GFP expressions were increased 4–5 times, while BX795 did not consistently and significantly improve transgene expression. We speculate that BX765 has unspecific effects at higher concentrations and attempted to achieve better STING inhibition by applying two inhibitors at maximal sub-toxic concentrations (Fig 2). However, adding 1 μM BX795 to MRT67307 adversely affected plasmid gene expression.

The goal of a plasmid transfection experiment is to examine the transgene's function in the host cell. In this case, the presence of a STING inhibitor is not ideal at the time of experiments. To address this issue, we determined 1) whether the transgene expression is sustained after the STING inhibitor is removed and 2) whether the inhibitor's effect on the STING pathway is reversible. In short, the answers to both questions are affirmative. Firstly, the transgene expression level in HAECs did not change 24 hours after the MRT67307 washout (Fig 3). The same experiment also showed that STING inhibition during transfection and the first 24 hours following transfection was critical for allowing transgene expression in HAECs. Secondly, the inhibition of the STING pathway by MRT67307 was highly reversible, as removing the inhibitor after a 24-hour treatment failed to enhance the transfection efficiency immediately following it (Fig 4).

Although MRT67307's effect is transient and reversible, a transfection experiment must be carefully designed. The STING pathway regulates not only inflammatory signaling but also cellular lipid and nucleic acid metabolism [35]. Both the introduction of plasmid DNA and the inhibitor inevitably disturb STING signaling, which can complicate the interpretation of the results in inflammatory and metabolic studies. We believe an appropriate control in a transfection experiment is an empty vector, as demonstrated in Fig 5. In this experiment, all cells were transfected with the same amount of plasmids, which were identical except for the transgene. All cells were also treated with MRT67307 at the same concentration and for the same duration. As a result, eNOS over-expression was confirmed by western blotting and immunostaining. Importantly, the over-expressed eNOS was enriched at its physiological compartments, i.e., plasma membrane and endoplasmic reticulum. As expected, the wild-type but not S1177A eNOS was phosphorylated at S1177 by VEGF treatment. This exemplifies how the STING-inhibitor-facilitated transfection can be used to study the transgene's biological function in a physiological context.

In addition to endothelial cells, STING inhibition by either MRT67307 or BX795 enhanced plasmid gene expression in HASMCs and RASMCs (Fig 6). Although the sensitivity to MRT67307 or BX795 varies between cell types, the best over-expression effects were achieved with 2–5 μM of STING inhibitors. The effective concentrations for primary endothelial and smooth muscle cells are higher than the 0.2 μM used by Fu Y for mouse embryonic fibroblasts [13], suggesting the presence of cytosolic plasmid DNA triggers stronger STING activation in primary vascular cells. Applying STING inhibitors at a higher concentration may also induce cytotoxicity. Therefore, while other hard-to-transfect primary cell types and cell lines may benefit from our transfection method, it is crucial to optimize the inhibitor concentration to maximize desired transgene expression with minimal toxicity when transfecting a different cell type.

In our transfection protocol, using more than 2.5 μg DNA /million HAECs did not result in more prominent plasmid gene expression (Figs 2 and 3). Instead, the transgene expression was inconsistent with 5 μg DNA/million HAECs between experiments, potentially due to the loss of cell viability. One exception is when cells were seeded at a lower density and allowed to proliferate for two days before transfection (Fig 4). In this case, the higher DNA load (5 μg/million cells) resulted in a 2-fold increase of the transgene expression compared to the lower DNA group (2.5 μg/million cells). Regardless of the DNA load, HAECs did not survive overnight transfection in our experience, which was not improved by STING inhibition. This suggests that the DNA-induced toxicity during transfection involves STING-independent mechanisms in HAECs. Apart from cGAS/STING activation, toll-like receptor 9 (TLR9) and absence in melanoma 2 (AIM2) initiate responses to cytosolic DNA [36]. It is worth investigating whether inhibiting these pathways improves cell viability, therefore allowing higher transfection efficiency.

Lastly, liposome-based transfection methods have been implemented to deliver plasmid DNA *in vivo* [25, 37, 38]. Interestingly, co-delivery siRNA targeting pro-inflammatory transcription factors, signal transducer and activator of transcription 1 (STAT1), and nuclear factor kappa B subunit 2 (NF-κB2) enhanced plasmid gene expression [37]. These transfection factors are responsible for interferon signaling and non-canonical NF-κB activation, which are regulated by STING signaling. As a future direction, it remains to be investigated whether the STING pathway inhibitors improve plasmid gene expression *in vivo*.

## Conclusions

We demonstrated that STING inhibition is an effective way to allow plasmid gene expression in difficult-to-transfect vascular endothelial and smooth muscle cells. In our transfection protocol, MRT67307 transiently suppresses STING activation, which is reversible after transgene is expressed. Our optimized transfection protocol provides an easy and cost-effective approach to expressing genes and studying their functions in primary vascular cells.

## Author Contributions

**Conceptualization:** Shuai Yuan.

**Data curation:** Shuai Yuan.

**Formal analysis:** Shuai Yuan.

**Funding acquisition:** Adam C. Straub.

**Investigation:** Shuai Yuan.

**Methodology:** Shuai Yuan.

**Supervision:** Adam C. Straub.

**Validation:** Adam C. Straub.

**Visualization:** Shuai Yuan.

**Writing – original draft:** Shuai Yuan.

**Writing – review & editing:** Shuai Yuan, Adam C. Straub.

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
