## [Decision Letter · Decision Letter 0]

5 Mar 2024

PONE-D-24-02106STING inhibition enables efficient plasmid-based gene expression in primary vascular cells: a simple and cost-effective transfection protocolPLOS ONE

Dear Dr. Yuan,

Thank you for submitting your manuscript to PLOS ONE. After careful consideration, we feel that it has merit but does not fully meet PLOS ONE’s publication criteria as it currently stands. Therefore, we invite you to submit a revised version of the manuscript that addresses the points raised during the review process.

We look forward to receiving your revised manuscript.

Kind regards,

Jérôme Robert, PhD

Academic Editor

PLOS ONE

Journal Requirements:

"Financial support for this work was provided by the National Institutes of Health R35 HL161177 (A.C.S.), National Institutes of Health R01 HL 153532 (A.C.S.), and American Heart Association Established Investigator Award 19EIA34770095 (A.C.S.)."

"Financial support for this work was provided by the National Institutes of Health (www.nih.gov) R35 HL161177 (A.C.S.), National Institutes of Health R01 HL 153532 (A.C.S.), and American Heart Association (www.heart.org) Established Investigator Award 19EIA34770095 (A.C.S.). Funders did not play any role in the study design, data collection and analysis, decision to publish, or preparation of the manuscript. "

Reviewers' comments:

Reviewer's Responses to Questions

**Comments to the Author**

1. Is the manuscript technically sound, and do the data support the conclusions?

Reviewer #1: Yes

Reviewer #2: Yes

2. Has the statistical analysis been performed appropriately and rigorously? 

Reviewer #1: Yes

Reviewer #2: Yes

3. Have the authors made all data underlying the findings in their manuscript fully available?

Reviewer #1: Yes

Reviewer #2: Yes

4. Is the manuscript presented in an intelligible fashion and written in standard English?

Reviewer #1: Yes

Reviewer #2: Yes

5. Review Comments to the Author

Reviewer #1: The ms has novelty and shows how Sting inhibitors can enhance transfection in cultured cells. My overall impression is that this is a well-conducted study. It appears a little brief in that only three cell lines were examined. I would have liked inclusion of primary cells or even in vivo transduction attempts to be included. The authors should discuss the in vivo applicability of their findings, not at least potential issues of toxicology of the Sting inhibitors.

Reviewer #2: In this paper, Yuan and Straub, show that inhibition of STING pathway could be beneficial to increase expression of transfected genes. This they prove in several ways, by direct treatments and also after chasing the drug away. They indeed use notably difficult to transfect cells (HAECs and HASMC for example) and comprehensively show that treatment with STING inhibitors increased the expression of the probed cDNA constructs. Even though there are few questions remaining. For example, one can only speculate that careful experimental design or result interpretation must be ensured, when opting for this strategy for metabolic studies. Similarly, there are variety of other ‘difficult to transfect cells’, for example, human monoploid HAP1 cells, extensively used for CRISPR knockout strategies. Authors may want to comment on the overall feasibility, not entirely focusing on vascular cells.

However, I support the publication of this article in PlasOne.

I have only a few minor comments/suggestions for the authors:

1. In the abstract, it would help to introduce the eNOS mutant.

2. In the methods part, the authors write, ‘The coding sequences of human cytochrome b5 reductase 4 (CYB5R4) and human nitric oxide synthase 3 (eNOS) were purchased from Transomic Technologies (Clone ID BC063294, BC025380).’ Please correct.

3. Correct units in the methods and elsewhere in figures, for example, u to µ (if that is what authors mean)

4. Authors write, ‘Membrane-bound primary antibodies were then probed by IRDye 680RD or 800CW secondary antibodies’. This statement may need correction.

5. ‘Western blotting shows 2-fold of WT or S1177A eNOS compared to the empty vector (EV) (Figure 5A).’ this is an open statement, and could be cleared whether the authors mean increase.

6. The discussion is too long. It may benefit from a fine cut. I, however, leave that to authors to decide.

6. PLOS authors have the option to publish the peer review history of their article (what does this mean?). If published, this will include your full peer review and any attached files.

Reviewer #1: No

Reviewer #2: No

---

## [Author Response · Author response to Decision Letter 0]

24 Apr 2024

Reviewer #1: 

The ms has novelty and shows how Sting inhibitors can enhance transfection in cultured cells. My overall impression is that this is a well-conducted study. It appears a little brief in that only three cell lines were examined. I would have liked inclusion of primary cells or even in vivo transduction attempts to be included. 

We appreciate the reviewer’s encouraging comments and agree that the current manuscript is concise and straightforward. However, our aim is to characterize and share a simple and cost-effective approach for transient gene expression in primary vascular cells such as endothelial cells and vascular smooth muscle cells. Importantly, all three cell types used in this study, human aortic endothelial cells, human aortic smooth muscle cells, and rat aortic smooth muscle cells, were primary cells commercially available from Lonza. We are sorry for the confusion and have emphasized this point in the Materials and Methods of the revised manuscript.

Regarding in vivo transduction, viral transduction using lentivirus or adeno-associated virus has been successfully implemented to express gene cargo in mouse vasculature (PMID: 26612671, PMID: 35571675, PMID: 35644115, PMID: 23535897). In these viral systems, the gene of interest can be effectively expressed in vascular endothelial and smooth muscle cells without STING inhibition. However, the more challenging part is to deliver the virus specifically and efficiently in vivo. Therefore, we believe that this is beyond the scope of the current manuscript.

The authors should discuss the in vivo applicability of their findings, not at least potential issues of toxicology of the Sting inhibitors.

We thank the reviewer for this suggestion. Indeed, lipofectamine and other liposome-based transfection methods can be used to deliver plasmid DNA or siRNA in vivo (PMID: 35879315, PMID: 31341189, PMID: 14585718, PMID: 32393755). In an interesting study, Zhu Y and colleagues demonstrated that co-delivery siRNA targeting pro-inflammatory transcription factors, STAT1 and NF-κB2, enhanced plasmid gene expression. These transfection factors are responsible for interferon signaling and non-canonical NF-κB activation. Conceptually, a similar effect can be achieved by inhibiting STING signaling, which activates both transcription factors. However, this approach is challenging in two ways. Firstly, tissue susceptibility to transfection is affected by the lipid property of the transfection reagent, and not all tissues effectively express the transgene (PMID: 32393755, PMID: 35879315). Therefore, it is difficult to target vascular endothelial or smooth muscle cells, which is the main focus of this manuscript. Secondly, although inhibitors used in this study have shown promising applications in autoimmune disease and virus-induced inflammation in vivo (PMID: 27353409, PMID: 37723181), the inhibitor’s tissue availability may differ from transfection complex, making it more difficult to express transgene in a specific tissue. One potential remedy is to encapsulate the inhibitor in or conjugate it to the liposome, which we intend to pursue as a future direction. We have included a brief discussion on this point in the revised manuscript.  

Reviewer #2: 

In this paper, Yuan and Straub, show that inhibition of STING pathway could be beneficial to increase expression of transfected genes. This they prove in several ways, by direct treatments and also after chasing the drug away. They indeed use notably difficult to transfect cells (HAECs and HASMC for example) and comprehensively show that treatment with STING inhibitors increased the expression of the probed cDNA constructs. Even though there are few questions remaining. For example, one can only speculate that careful experimental design or result interpretation must be ensured, when opting for this strategy for metabolic studies. 

We are grateful for the reviewer’s positive comments. We agree that plasmid transfection with STING inhibitors requires careful experimental design and result interpretation. We have extended the discussion on experimental design and the optimal control in a transfection experiment in the revised manuscript. 

Similarly, there are variety of other ‘difficult to transfect cells’, for example, human monoploid HAP1 cells, extensively used for CRISPR knockout strategies. Authors may want to comment on the overall feasibility, not entirely focusing on vascular cells.

Indeed, there are other primary cell types and cell lines that are resistant to plasmid gene expression. However, it is not practical to exhaust those choices. In addition to the cells used in the manuscript, we also tested our method on primary human pulmonary microvascular endothelial cells (HPMVEC), ovarian cancer cells (OVCAR4), and mouse bone marrow-derived mononuclear cells (BMDM). While MRT67307 enhanced plasmid gene expression in HPMVEC and OVCAR4, it was ineffective in BMDM. Interestingly, gene silencing can be achieved by lipofectamine-based siRNA transfection in BMDM, suggesting liposome delivery was not the problem. It may be challenging to fully suppress STING activation in some cell types, or an alternative mechanism is regulating plasmid gene expression. However, this is only our speculation, and we are not confident to elaborate on it in the discussion. 

Overall, we believe it is possible to extend our transfection method to other hard-to-transfect cells, but experimental conditions need to be tested and optimized for individual cell types. This point is now emphasized in the discussion. 

I have only a few minor comments/suggestions for the authors:

1. In the abstract, it would help to introduce the eNOS mutant.

A brief introduction on eNOS S1177A mutant has been added to the abstract. 

2. In the methods part, the authors write, ‘The coding sequences of human cytochrome b5 reductase 4 (CYB5R4) and human nitric oxide synthase 3 (eNOS) were purchased from Transomic Technologies (Clone ID BC063294, BC025380).’ Please correct.

This sentence has been corrected.

3. Correct units in the methods and elsewhere in figures, for example, u to µ (if that is what authors mean)

We have corrected this typo in the methods and figures. 

4. Authors write, ‘Membrane-bound primary antibodies were then probed by IRDye 680RD or 800CW secondary antibodies’. This statement may need correction.

We apologize for the confusion. This section has been modified with more details. 

5. ‘Western blotting shows 2-fold of WT or S1177A eNOS compared to the empty vector (EV) (Figure 5A).’ this is an open statement, and could be cleared whether the authors mean increase.

This statement has been clarified. 

6. The discussion is too long. It may benefit from a fine cut. I, however, leave that to authors to decide.

We thank the reviewer for this suggestion and agree that the discussion is long, considering our findings are straightforward. However, we believe the audience can benefit from a thorough discussion that presents the advantages and limitations of our transfection method and allows them to adopt and improve it in their own studies.

---

## [Editor Report · Decision Letter 1]

25 Apr 2024

STING inhibition enables efficient plasmid-based gene expression in primary vascular cells: a simple and cost-effective transfection protocol

PONE-D-24-02106R1

Dear Dr. Yuan,

We’re pleased to inform you that your manuscript has been judged scientifically suitable for publication and will be formally accepted for publication once it meets all outstanding technical requirements.

Kind regards,

Jérôme Robert, PhD

Academic Editor

PLOS ONE

Additional Editor Comments (optional):

Please ensure you deposited the data as mentioned ((10.6084/m9.figshare.24982932). As of today it is not accessible.
---

## [Editor Report · Acceptance letter]

1 Jul 2024

PONE-D-24-02106R1 

PLOS ONE

Dear Dr. Yuan, 

I'm pleased to inform you that your manuscript has been deemed suitable for publication in PLOS ONE. Congratulations! Your manuscript is now being handed over to our production team.

Kind regards, 

on behalf of

Dr. Jérôme Robert 

Academic Editor

PLOS ONE